# INVESTIGATION OF NUMERICAL DIFFUSION IN AERODYNAMIC FLOW SIMULATIONS WITH PHYSICS INFORMED NEURAL NETWORKS

**Alok Warey**[*]**, Taeyoung Han & Shailendra Kaushik**
General Motors Global Research and Development
Warren, MI 48092, USA
`alok.warey@gm.com`

## ABSTRACT

Computational Fluid Dynamics (CFD) simulations are used for many air flow simulations including road vehicle aerodynamics. Numerical diffusion occurs when local flow direction is not aligned with the mesh lines and when there is a non-zero gradient of the dependent variable in the direction normal to the streamline direction. It has been observed that typical numerical discretization schemes for the Navier-Stokes equations such as first order upwinding produce very accurate solutions without numerical diffusion when the mesh is aligned with the streamline direction. On the other hand, numerical diffusion is maximized when the streamline direction is at an angle of 45° relative to the mesh line. Numerical diffusion can be reduced by mesh refinements such as aligning mesh lines along the local flow direction or by introducing higher order numerical schemes, which may introduce potential numerical instability or additional computational cost. A few test cases of a simple steady-state incompressible and inviscid air flow convection problem were used to investigate whether numerical diffusion occurs when using Physics Informed Neural Networks (PINNs) that rely on automatic differentiation as opposed to numerical techniques used in traditional CFD solvers. Numerical diffusion was not observed when PINNs were used to solve the partial differential equation (PDE) for the simple convection problem irrespective of flow angle. The PINN correctly simulated the streamwise upwinding, which has great potential to improve the accuracy of Navier-Stokes solvers.

## 1 INTRODUCTION

Understanding flow phenomena and, especially, how aerodynamic forces are influenced by changes in vehicle body shape, are very important to improve vehicle aerodynamic performance particularly for low drag shapes. One of the goals of vehicle aerodynamics simulations is to predict the influence of changes in body shape on the flow field and, thereby, on drag and lift forces. Due to numerical diffusion (numerical inaccuracy) of current discretization schemes for the convection terms, accurate prediction of drag and lift forces with a reasonable number of mesh points has been challenging. Therefore, the tendency is to increase the number of mesh points to improve numerical accuracy. However, the demand for mesh refinement is very high and typically a large number of mesh points (more than 100 million for external aerodynamics simulations) are needed. Typical external aerodynamic simulations with large number of mesh points tend to be very inefficient due to slow turnaround times (Leer, 1985; Karadimou & Markatos, 2018; Patel et al., 1985; 1988; Han et al., 2011) .

## 2 NUMERICAL DIFFUSION

Numerical diffusion (or inaccuracy) occurs when local flow direction is not aligned with the mesh lines and when there is a non-zero gradient of the dependent variable in the direction normal to the

---

[*]Corresponding author

streamline direction. It has been observed that a typical numerical discretization scheme (first order upwinding) for the Navier-Stokes equations produces very accurate solutions without numerical diffusion when the mesh lines are aligned with the streamline direction. On the other hand, numerical diffusion is maximized when the streamline direction makes an angle of 45° with respect to the mesh line. Numerical diffusion can be reduced by mesh refinements or by introducing higher order numerical schemes, which introduce potential numerical instability or additional computational cost (Leer, 1985; Karadimou & Markatos, 2018; Patel et al., 1985; 1988; Han et al., 2011).

To explain the effects of numerical diffusion, consider a simple steady-state convection problem for incompressible and inviscid air flow given by the energy equation below:

$$u\frac{\partial T}{\partial x} + v\frac{\partial T}{\partial y} = 0 \tag{1}$$

where u is the air velocity in the x direction, v is the air velocity in the y direction and T is the air temperature. The problem setup with a quadrilateral mesh is shown in Figure 1. A uniform velocity field boundary condition (BC) of 1.0 m/s was specified at the inlet boundary at different angles relative to the mesh line. A non-dimensional air temperature of 1.0 was specified above the dividing streamline and 0.0 below the dividing streamline. With no viscous diffusion (inviscid air flow), no mixing layer should form and the exact solution for this equation should have a temperature discontinuity across the dividing streamline. If numerical diffusion is present in the calculations, then the numerical solution produces a mixing layer in the downstream flow. The amount of numerical diffusion introduced by the numerical scheme can be estimated by examining the size of the mixing zone across the dividing streamline. Effect of numerical diffusion for one of the most popular numerical schemes in CFD, the first order upwinding scheme, was investigated for the following two cases:

Case 1: Mesh lines are aligned with the flow direction
Case 2: Flow angle relative to the mesh lines is 45°

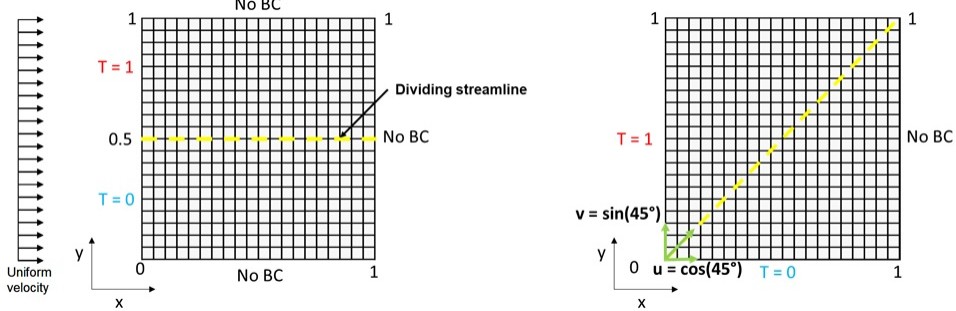

Figure 1: Problem setup for Case 1 and Case 2.

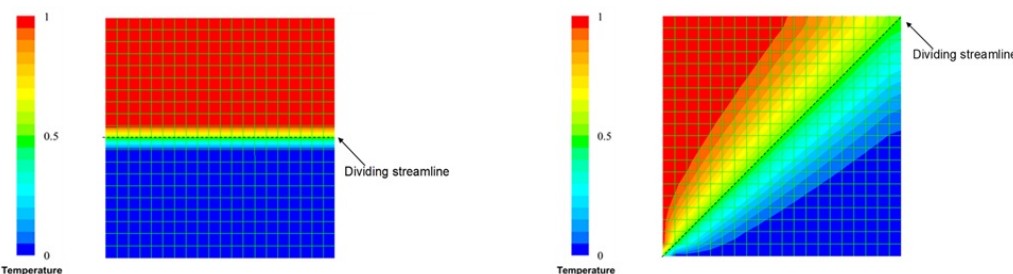

Figure 2: Computed temperature field for Case 1 and Case 2.

Figure 2 shows the computed temperature field for both cases. For Case 1, where the mesh lines are aligned with the flow direction, no mixing layer was observed, and a sharp temperature discontinuity

persisted in the streamwise direction. For Case 2, significant numerical diffusion was introduced in the solution when the flow direction was at 45° with respect to the mesh lines for the quadrilateral mesh. This is because no upstream flow information is readily available in the diagonal direction for the quadrilateral mesh. A simple approach to provide the upstream information in the diagonal direction is to introduce mesh lines by splitting the quadrilateral elements in the diagonal direction. A three-dimensional (3D) boundary layer on the vehicle surface has a cross-flow velocity component in the viscous flow regions. Therefore, aligning the mesh lines along the local flow direction within the boundary layer is not a trivial exercise. However, the effect of numerical diffusion becomes less severe as viscous diffusion starts dominating in the boundary layer.

## 3 PHYSICS INFORMED NEURAL NETWORKS

Neural networks can be used as a method for efficiently solving difficult partial differential equations (PDEs) that are commonly encountered in science and engineering - Physics-Informed Neural Networks (PINNs). Physics is explicitly imposed by constraining the output of conventional neural network architectures. PINNs provide a mesh free alternative compared to traditional numerical methods and the potential to significantly reduce computational cost (Lu et al., 2020; Raissi et al., 2017a;b; 2019). A schematic of a physics informed neural network for solving equation 1 above is shown in Figure 3.

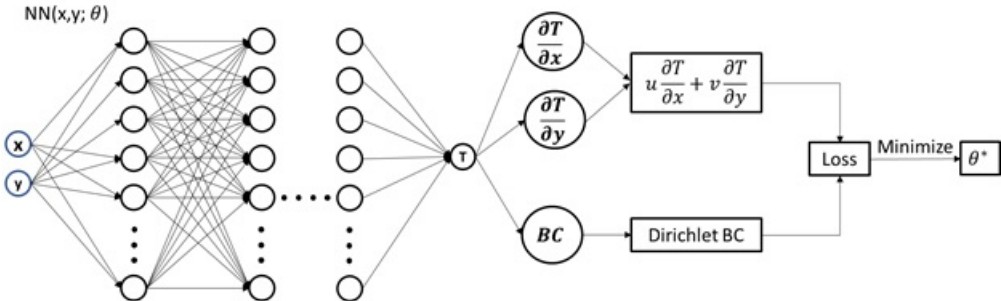

Figure 3: Schematic of the PINN used to solve equation 1.

Cases 1 and 2 described above were used to investigate whether numerical diffusion occurs when using PINNs that rely on automatic differentiation as opposed to finite differencing. The PINN was implemented with the Python library, DeepXDE (Lu et al., 2020), which is designed to serve both as an education as well as a research tool for solving problems in computational science and engineering. A uniform grid of 5041 points in the domain (71x71 grid) and 1000 points along the boundary were used for training the PINN. Predictions with the trained network were done on a 200x200 grid. Figure 4 shows the predicted temperature field on a 200x200 grid by a trained PINN for Case 1: flow angle = 0° and Case 2: flow angle = 45°. For Case 1, no mixing layer was observed, and a sharp temperature discontinuity persisted in the streamwise direction similar to the CFD solution. For Case 2, no numerical diffusion was observed unlike the first order upwinding numerical scheme.

Comparison of the temperature profiles at x = 0.5 between the PINN and first and second order upwinding schemes for various flow angles are shown in Figures 5 and 6. Severe errors were introduced when the flow angle was not aligned with mesh lines. Practically, it is impossible to align the mesh lines to the streamlines in three-dimensional (3D) flows. To reduce numerical diffusion, higher order numerical schemes are typically applied. However, these schemes are not bounded and can be potentially unstable. Higher order numerical schemes often introduce artificial diffusion to stabilize the solution. The PINN approach does not need an artificial viscosity to stabilize the solution since there are no truncation errors in the gradient calculations. The real advantage of a PINN seems to be that it naturally adapts the streamwise upwinding, which is the basic characteristic of pure convection. For the test problem with 30° flow angle, there is no upstream temperature information for a uniform point distribution in the computational domain, although, for 0° and 45° flow angles, upstream flow information is available from points upstream. The second order upwinding scheme

tends to reduce the numerical diffusion as shown in Figure 6, however, the PINN demonstrated far better accuracy for various flow angles.

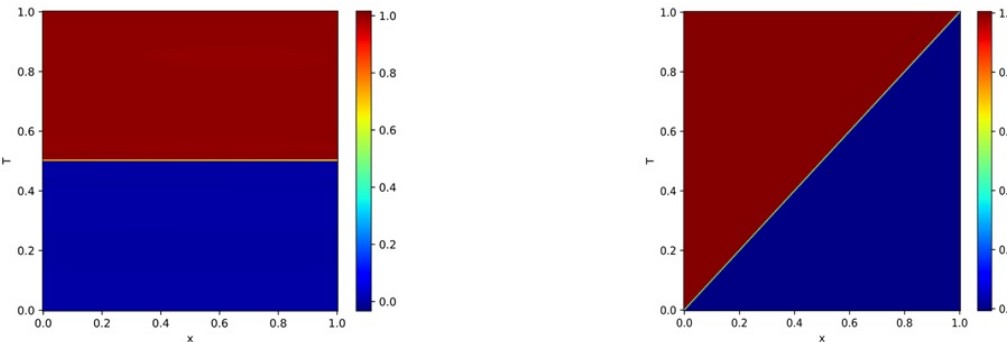

Figure 4: Predicted temperature field on a 200x200 grid by a trained PINN for Case 1: flow angle = 0° and Case 2: flow angle = 45°.

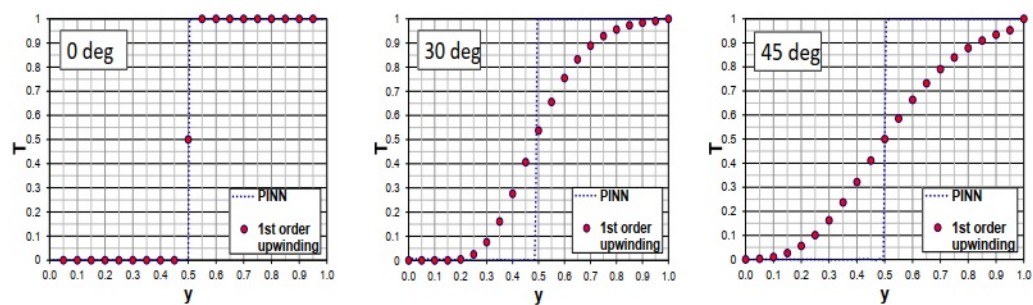

Figure 5: Comparison of the temperature profile at x = 0.5 between the PINN and 1st order upwinding scheme for various flow angles = 0°, 30°, 45°.

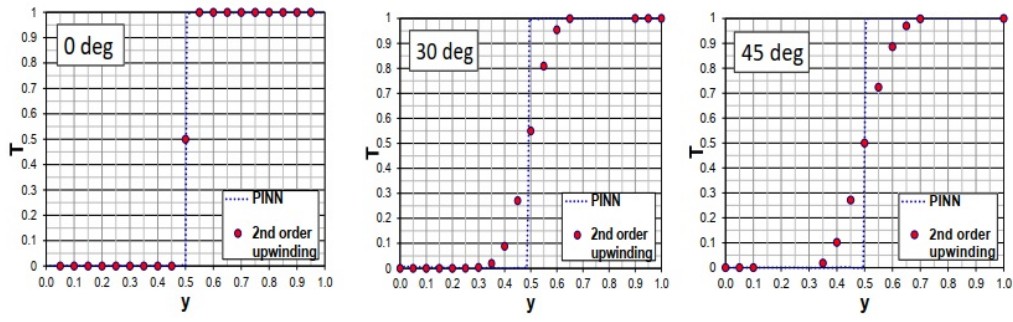

Figure 6: Comparison of the temperature profile at x = 0.5 between the PINN and 2nd order upwinding scheme for various flow angles = 0°, 30°, 45°.

## 4 SUMMARY

A numerical scheme must satisfy the necessary criteria for a successful solution of convection-diffusion formulations. Numerical diffusion (or inaccuracy) occurs with typical numerical schemes for the Navier-Stokes equations such as first order upwinding. Numerical diffusion can be reduced by mesh refinements such as aligning mesh lines along the local flow direction or by introducing higher order numerical schemes. A few test cases of a simple steady-state convection problem for incompressible and inviscid air flow were used to investigate whether numerical diffusion occurs when using Physics Informed Neural Networks (PINNs) that rely on automatic differentiation as opposed to numerical techniques used in traditional CFD simulations. Numerical diffusion was not observed when PINNs were used to solve the partial differential equation (PDE) for a simple steady-state convection problem for incompressible and inviscid air flow irrespective of flow angle. The PINN correctly simulated the streamwise upwinding, which has great potential to improve the accuracy of Navier-Stokes solvers.

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
