# OpenReview forum: "Investigation of Numerical Diffusion in Aerodynamic Flow Simulations with Physics Informed Neural Networks"
_ICLR.cc/2024/Workshop/AI4DiffEqtnsInSci — AI4DiffEqtnsInSci @ ICLR 2024 Poster_

### Official Review · Reviewer_YwEh · 2024-02-23
**Interesting case for PINNs over traditional solvers**

**Rating:** 8
**Confidence:** 4

**Review:**

The authors make an interesting case for using PINNs to solve PDEs over traditional semi-discretization methods and solvers. They provide a simple example of a steady-state PDE that experiences strong numerical diffusion for a case when the gridlines are at a 45-degree angle to the convection velocity. The authors convincingly show that solving the same PDEs with PINNs does not produce any numerical diffusion.

Pros:
- The work makes a really nice case for using PINNs to solve PDEs over traditional methods. This could be a convincing argument for sceptics why PINNs may be useful in their work

Cons:
- The authors do not give a lot of details about how they solved the steady state problem with classical methods
-  While the contribution makes a novel argument for why to use PINNs, the method presented itself is not novel

---

### Meta-Review · Area_Chair_4myD · 2024-03-01

**Recommendation:** Accept (Poster)

**Metareview:**

The paper addresses the common phenomenon of numerical diffusivity that occurs in conventional numerical solvers and explored whether the same thing could happen in PINNs framework. One confusion I have is neural networks through backpropagation automates the differentiation w.r.t input and I'm not sure how one can define the concept of numerical diffusion for neural networks. however, the results show that PINNs results do not look as diffusive as a coarse grid with upwind discretization scheme, which could be a promising advantage behind ML-based models.

---

### Decision · Program_Chairs · 2024-03-01

Accept (Poster)